# Design of an Einzel Lens with Square Cross-Section

**Michał Krysztof**

Department of Microsystems, Faculty of Electronics, Photonics and Microsystems, Wroclaw University of Science and Technology, ul. Z. Janiszewskiego 11/17, 50-372 Wroclaw, Poland; michal.krysztof@pwr.edu.pl

**Abstract:** In this paper, the results of modeling and simulation of a microcolumn are presented. The microcolumn is part of a developed miniature MEMS electron microscope equipped with a miniature MEMS high-vacuum micropump. Such an arrangement makes this device the first stand-alone miniature electron-optical device to operate without an external high-vacuum chamber. Before such a device can be fabricated, research on particular elements must be carried out to determine the working principles of the device. The results of the calculations described in this article help us to understand the work of a microcolumn with square holes in the electrodes. The formation of an electron beam spot at the anode is discussed. Further calculations and results show the dependence of the Einzel lens size on the electron beam spot diameter, electron beam current, and microcolumn focusing voltage. The results are used to define the optimal design of the developed MEMS electron microscope.

**Keywords:** microcolumn; Einzel lens; focusing of electron beam; MEMS; miniaturization

## 1. Introduction

Since the end of the 1980s, research has been carried out to fabricate miniature electron optical columns called microcolumns. Such devices could be used to develop a small and inexpensive electron microscope or electron lithography tool. Moreover, the microcolumns could be connected in a matrix to form a multicolumn device, which could increase the throughput of the devices. Due to the size of the microcolumn, classical magnetic lenses could not be used. Instead, electrostatic lenses were used. Electrostatic lenses have higher aberrations. However, with miniaturization, the aberrations become smaller.

In 1996, Chang summarized the information on microcolumns from his previous studies [1]. According to Chang, to develop a microcolumn with good parameters, small apertures < 100 μm and a small distance between electrodes < 500 μm must be used. However, such an arrangement limited the electron beam energy that could be used with such a microcolumn to 1 keV. This was due to a possible electrical breakdown between the electrodes.

Throughout the years, different approaches were proposed to make microcolumns using different techniques and different materials [2–9]. Usually, for the fabrication of the devices, microengineering methods were used, i.e., microfabrication of silicon and glass and anodic bonding for joining the elements. In almost all cases, the theory of Chang was implemented and microcolumns were fabricated with small apertures and small distances between electrodes. The results of these investigations were promising. It was possible to fabricate the microcolumn with very good imaging parameters. However, the proposed devices worked inside a high vacuum chamber, which is a drawback for the miniaturized electron beam device, which still needs a large vacuum housing and pumping system.

In 2014 [10,11], a new microcolumn solution was proposed that consists of all the parts needed to fabricate a miniature scanning electron microscope (Figure 1). The most important innovation of this instrument is that it is equipped with a miniature MEMS

high vacuum micropump, which ensures a high vacuum (up to $10^{-7}$ mbar) within the microdevice [12]. The device is designed as a stand-alone microscope that can operate without external high-vacuum devices. To achieve such a structure, a change in the microcolumn design proposed by Chang was made. In the proposed solution, silicon electrodes and glass spacers play the role of vacuum housing; therefore, the structure of the devices has to be robust. In addition, the electron beam generated inside the microcolumn needs to be transported through the thin silicon nitride membrane to the sample. This involves higher electron beam energies > 1000 eV. To ensure such conditions, thick glass spacers (1.1 mm) were used to eliminate electrical breakdown between the electrodes. It was confirmed that such a thickness of the glass is suitable for withstanding a 6 kV voltage difference on the electrodes. A novelty in this solution, from the beginning of the project, is the square holes in the electrodes that build the microcolumn. This is chosen because the fabrication of the microcolumn should be as simple as possible. Fabrication of circular holes requires RIE (reactive ion etching) or DRIE (deep reactive ion etching) processes, which are expensive and utilize expensive tools. However, square holes can be made using an anisotropic silicon etching, which is not expensive and is a precise silicon micromachining technique. Throughout the years, particular elements of the designed microscope were fabricated and tested: emitters [13,14], electrostatic lenses [15], membranes [16], deflector systems [17], and the results show that it is possible to fabricate a standalone miniature MEMS microscope.

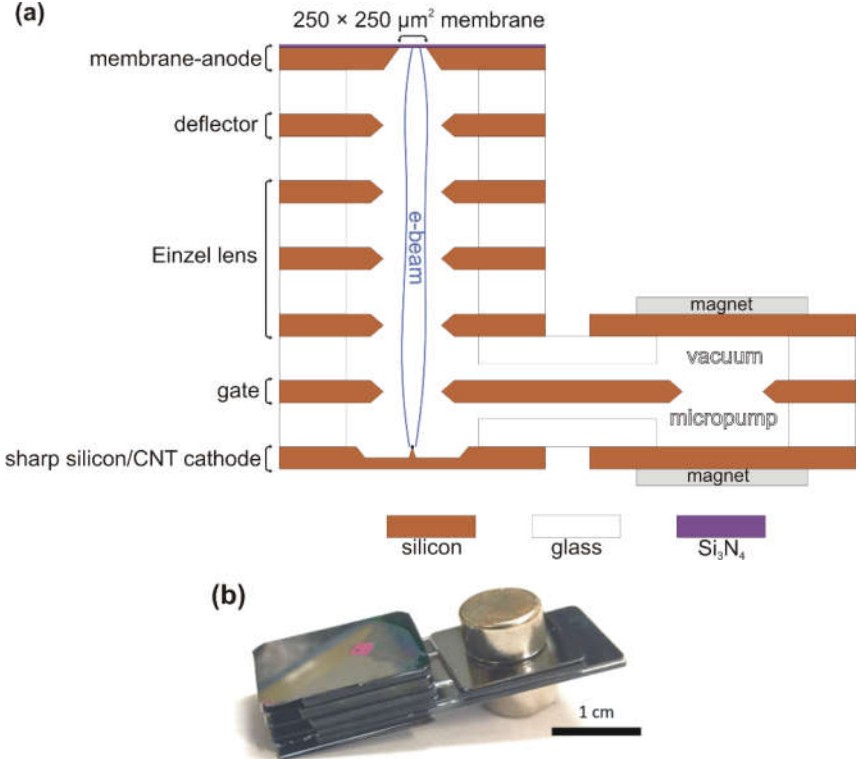

**Figure 1.** Concept of the complete MEMS electron microscope: (**a**) scheme; (**b**) photo of the assembled structure without deflector.

Although the circular Einzel lenses are most commonly used for the fabrication of microcolumns, as their axial symmetry introduces little distortion to the electric field of the lens, this article presents the results of simulations concerning the design of the square Einzel lens. Several models were created and calculated using the SIMION 3D v.7.0 software. All models resolve the physical models of the microcolumns fabricated and tested

in our laboratory. The study aims to understand the physics of the square Einzel lens and to see if it is suitable for miniature MEMS electron microscope fabrication. In addition, the results will help optimize the structure of the microcolumn. The microcolumn fabricated with square holes in electrodes is designed not to compete with conventional microcolumns in terms of electron beam parameters but rather as a cheap and easy-to-make alternative.

## 2. Modeling Using a Parallel Electron Beam

The simulations concern only the Einzel lens, that is why the simulated model contains only six electrodes (Figure 2): a cathode with a CNT layer for electron emission (1), a gate electrode (2), three Einzel lens electrodes (3, 4, 5), and an anode (6). At first, the cathode and anode are flat electrodes. At the cathode, a 20 μm thick CNT layer is defined as 1 mm × 1 mm. The electrodes in the middle have square holes. The gate hole size was marked as $a_G$ because in some experiments this size is different from the size of Einzel lens electrodes ($a$). In reality, the holes are etched in 10 M KOH solution, which makes the holes not straight but etched at an angle of ~54 deg. Due to the resolution of the model and the fact that the 3D model in SIMION is built from small cubes, the angle is changed to 45 degrees. All electrodes are separated by a distance of $h$ = 1.1 mm, which corresponds to the thickness of the glass used as a separator.

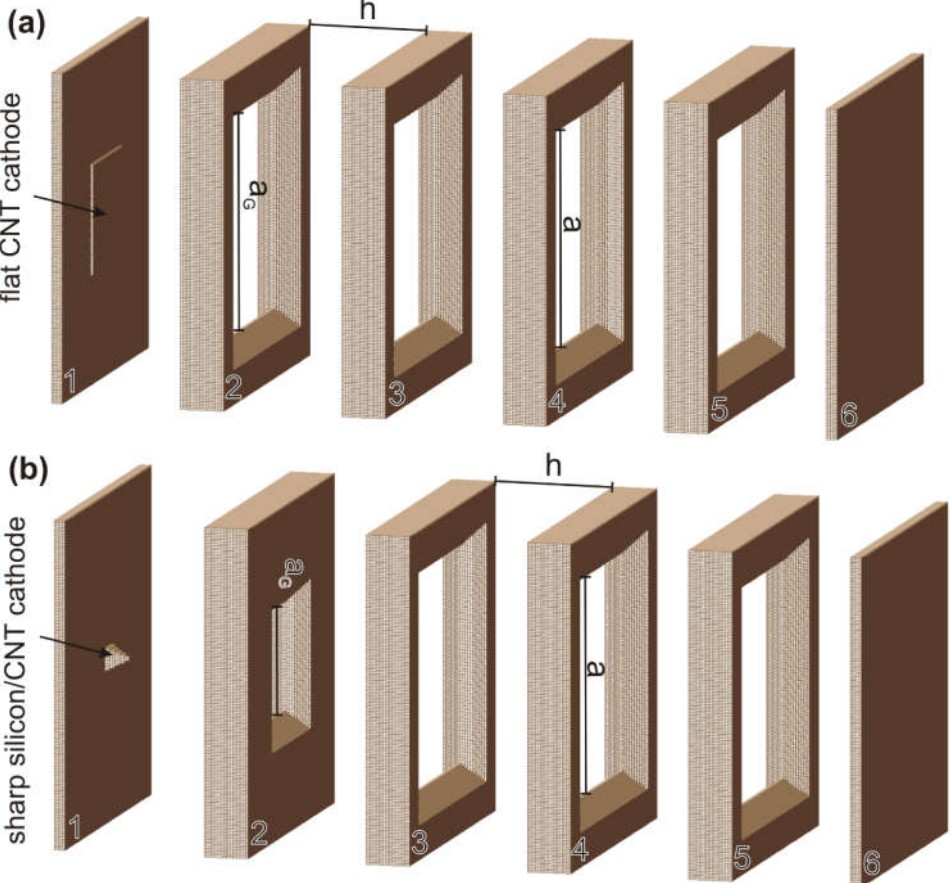

**Figure 2.** Examples of microcolumn model applied in SIMION 3D v.7.0 software: (**a**) flat CNT cathode model $a = a_G = 2$ mm; (**b**) sharp silicon/CNT cathode model $a = 2$ mm, $a_G = 1$ mm (1—cathode, 2—gate, 3, 4, 5—Einzel lens, 6—anode).

In the first experiment, a theoretical electron beam was defined consisting of 40,401 electrons uniformly distributed on the 1 mm × 1 mm surface. All electrons start perpendicularly to the surface of the emission layer and have the same starting energy of 1 eV. Such an ideal parallel beam was used to investigate the ideal parameters of the square Einzel lens. For the first experiment, the model with size $a = 1$ mm was chosen. The voltage at the cathode was set at $U_C = -2000$ V and at the gate $U_G = -1000$ V. These values were constant throughout the experiment. The anode and two external Einzel lens electrodes (electrodes 3 and 5) were in the ground state ($U_A = U_3 = U_5 = 0$ V). Only the voltage of the middle Einzel lens electrode (electrode 4), called the focusing electrode, was changed. At first, the voltage was set at $U_F = -1000$ V and changed every $-100$ V until no electrons were observed at the anode. Next, a smaller interval was taken and the simulation was performed every 10 V to find the best focus voltage ($U_F$). During the simulation, the electron coordinates at the anode were collected, and from that, the focusing of the electron beam was calculated (Figure 3).

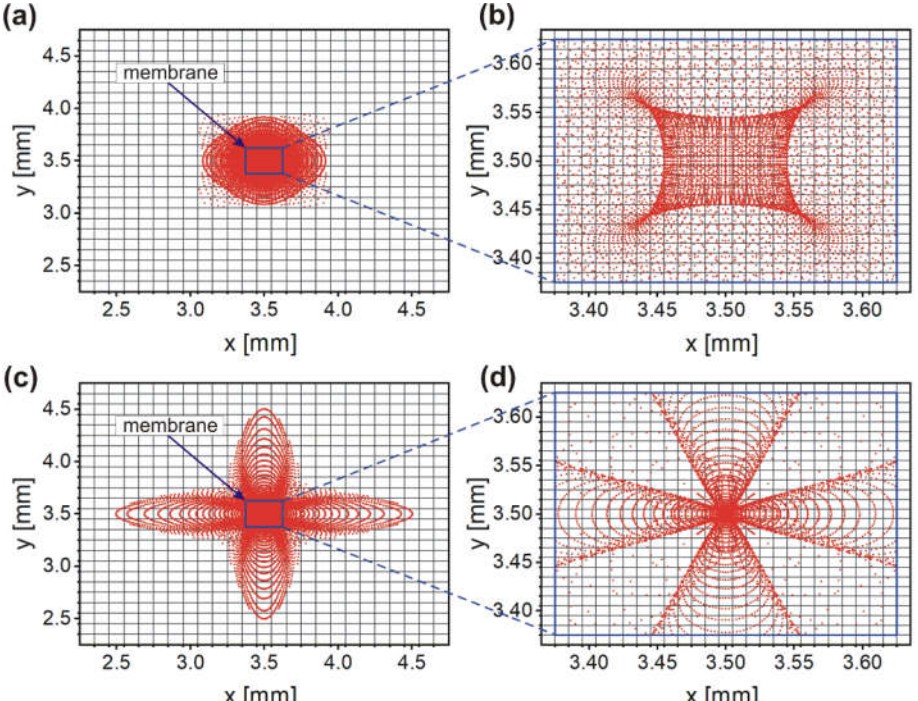

**Figure 3.** Electron beam coordinates recorded for 1 mm Einzel lens for: (**a**) $U_F = -1700$ V—anode; (**b**) $U_F = -1700$ V—membrane; (**c**) $U_F = -1840$ V—anode; (**d**) $U_F = -1860$ V—membrane.

The surface of the anode was divided into small squares with an edge of 0.1 mm (Figure 3a,c, dark grey lines). Then, from the electron coordinates on the anode, the electron count was made in every square, which translates to a matrix of 49 × 49 that illustrates the electron intensity on the surface of the anode with a resolution of 0.1 mm. Using those matrices for every voltage, 3D plots and heat maps were made to observe the formation of the electron beam peak at the anode (Figure 4).

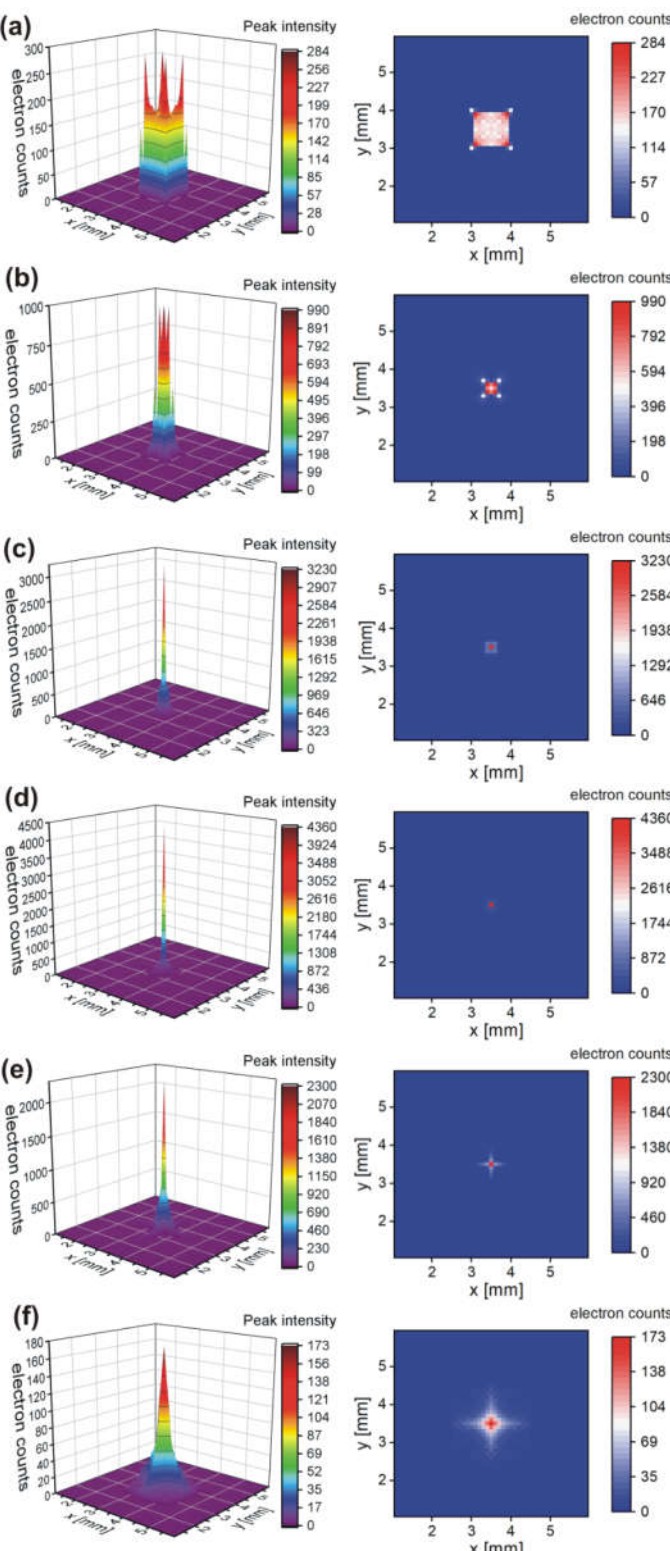

**Figure 4.** Three-dimensional plots and heat maps of the beam spot at the anode for different focusing voltages: (**a**) $U_F = -1000$ V; (**b**) $U_F = -1500$ V; (**c**) $U_F = -1700$ V; (**d**) $U_F = -1840$ V; (**e**) $U_F = -1900$ V; (**f**) $U_F = -2100$ V.

Looking at the 3D plots and heat maps, a change in the shape of the electron beam spot is visible. When −1000 V is applied to the focusing electrode, the spot is a square with the edge of appr. 1 mm. However, the concentration of electrons is observed in the corners of the square (Figure 4a). When a higher voltage was applied ($U_F$ = −1500 V), the edges of the square bend and the concentration of electrons in the corners increased (Figure 4b). The spot is similar to the letter "X". After reaching a certain voltage ($U_F$ = −1700 V, for $a$ = 1 mm), the four peaks are so close together that all electrons concentrate in the center square defined in the matrix (Figure 4c). The peak increases with increasing voltage until a maximum is reached (Figure 4d). After that, the shape of the spot changes from the letter "X" to a plus sign "+", and the electron peak decreases (Figure 4e,f). The voltage with the highest electron concentration in the middle is the best focus voltage for given electrode dimensions ($a$ = 1 mm). When the electron peak becomes singular, additional calculations were made to estimate the thickness of the peak. Since the shape of the electrodes is square and the shape of the electron beam spot is symmetric to the $x$ and $y$ axes, we calculated the profiles of the peak in the $x$ and $y$ directions and calculated the FWHM (full width half maximum) parameter for the peaks observed in the 3D plots. Then the average of the two values was defined as the diameter of the beam spot for a given model $D$. For $a$ = 1 mm and 2 keV electron beam, the focus voltage was $U_F$ = −1840 V and the diameter of the beam $D$ = 0.106 mm. The calculated beam diameter reached the resolution limit of 0.1 mm and showed that the focusing power of the square Einzel lens is very good, knowing that the emitter size is 1mm × 1 mm. Rough calculations are made to find the best focus voltage.

The developed miniature MEMS electron microscope is equipped with a thin silicon nitride membrane, which is used as an electron transparent window. The size of the membrane is 250 μm × 250 μm. The simulation showed that the electrons emitted from the 1×1 mm$^2$ emitter can be focused at least to $D$ = 0.1 mm. However, the rest of the electrons can cover an area of about 5 mm × 5 mm. Since the silicon nitride membrane has a size of 250 μm × 250 μm, it is best to include in the calculations only the electrons that hit the anode in the field covered by the membrane (Figure 3a,c, blue square). The rest of the electrons are not playing an important role in the further imaging process because they are screened by the silicon anode. Therefore, detailed calculations were performed using a 27 × 27 matrix with squares with an edge of 0.01 mm (Figure 3b,d, dark gray lines). The resolution was set as a compromise between the speed and accuracy of the calculations. With such a resolution, it is possible to see what is happening inside the peak that was calculated before. The distribution of electrons of the same electron beam but in a smaller area can be observed (Figure 5). Calculations showed that the best focusing voltage is slightly higher $U_F$ = −1860 V, which gives the electron beam spot size of $D$ = 0.0106 mm. This value is also at the edge of the resolution of the calculation method; however, those calculations were made to choose the proper method for calculating the electron beam parameters.

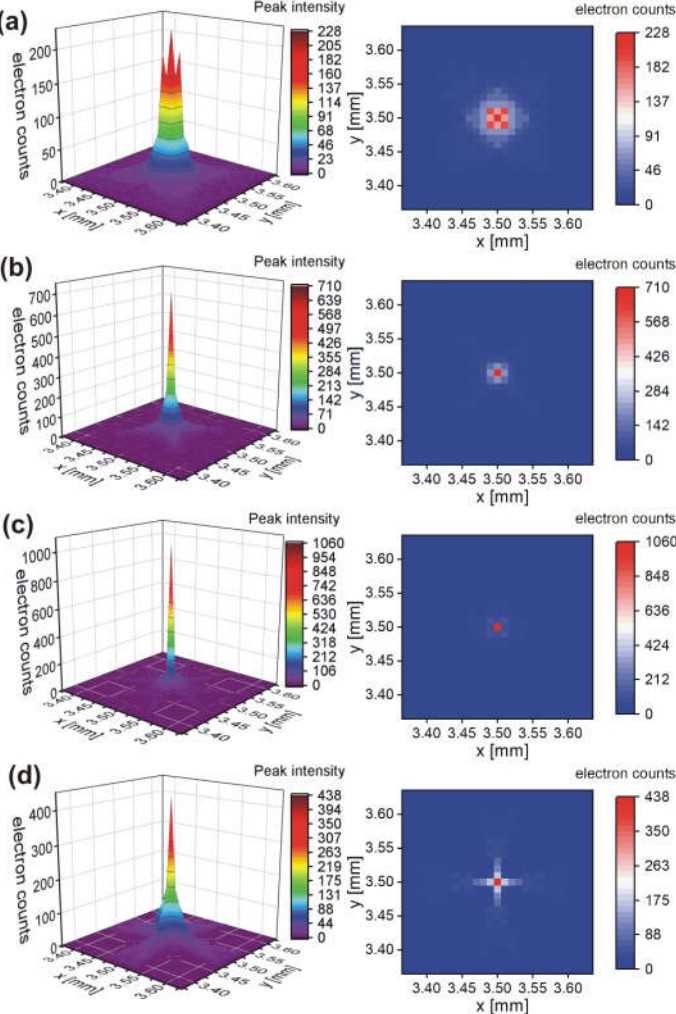

**Figure 5.** Three-dimensional plots and heat maps of the beam spot on the membrane for different focal voltages: (**a**) $U_F = -1820$ V; (**b**) $U_F = -1840$ V; (**c**) $U_F = -1860$ V; (**d**) $U_F = -1880$ V.

## 3. Modeling Using Realistic Electron Beam

### 3.1. Flat CNT Cathode

For the next experiments, a change in the primary beam was made. Instead of an ideal parallel beam, a realistic beam was defined. 50,000 electrons were randomly and uniformly spread over a 1 mm × 1 mm emission field. Moreover, the starting angles of the electrons were also randomized, as well as the starting energy, which was randomly chosen from 1 to 5 eV. Such starting conditions were more similar to the real emitter that we used in the first experiments with focusing on the electron beam [15]. A flat silicon cathode was used, on which a 1 mm × 1 mm layer of CNT was electrophoretically deposited. Because the surface of the layer is not uniform, the emission from such a layer is also not uniform. The energy spread is consistent with the electron beam energy spread emitted from CNTs. With such a defined electron beam, two experiments were conducted.

In the first experiment (Exp. 1), the gate size and the Einzel lens size were changed simultaneously from $a = a_G = 1$ to 3 mm every 0.5 mm. A model with a hole size of 2 mm is presented in Figure 2a. As stated in Chapter 2, the voltages at all electrodes were kept constant ($U_C = -2000$ V, $U_G = -1000$ V, $U_A = U_3 = U_5 = 0$V) and only the focusing electrode voltage was changed. Knowing the best focus voltage for the ideal beam, simulations were

performed close to this value every 10 V and the calculations of $D$ were made. The calculations were repeated for a larger Einzel lens size. The results are presented in Figure 6 (plot 1).

For comparison, a second experiment (Exp. 2) was conducted in which the gate electrode size was kept constant ($a_G$ = 1 mm) and the size of the Einzel lens electrodes was changed (from $a$ = 1 to 3 mm). Such an electrode arrangement with a 2 mm Einzel lens is presented in Figure 2b, but in this experiment the cathode was flat. Similar simulations and estimations of the smallest beam diameter were carried out for each model. The results are superimposed on the previous results in Figure 6 (plot 2).

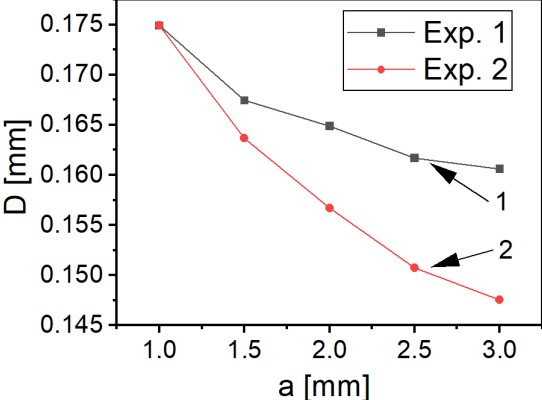

**Figure 6.** Electron beam diameter as a function of Einzel lens size (plot 1—data for the first experiment, plot 2—data for the second experiment).

For both experiments, more accurate calculations for the membrane area were made to see how the beam intensity is distributed on the membrane. However, for the randomly defined beam, no high peak is observed at the membrane surface (Figure 7).

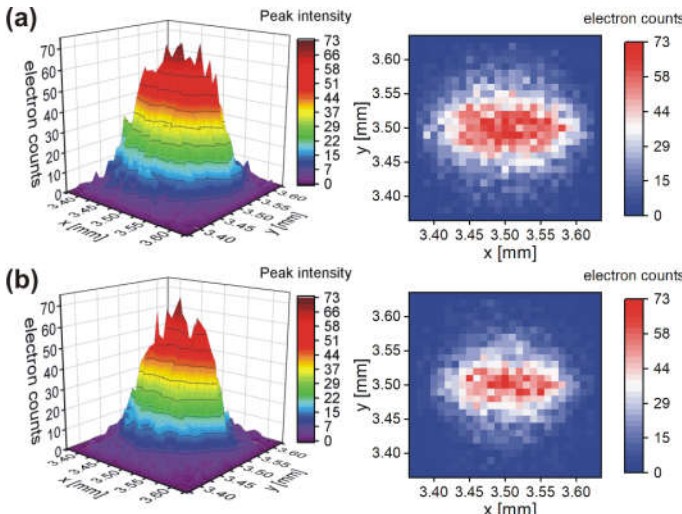

**Figure 7.** Three-dimensional plots and heat maps for the best focus voltages on the membrane: (**a**) first experiment; (**b**) second experiment.

What is interesting when analyzing Figure 6 is that the size of the electron beam decreases with the increasing size of the holes. It is not a great change (14.3 μm for the first experiment and 27.4 μm for the second experiment), however, if the best model for focusing the electron beam using a square Einzel lens must be chosen, the 3 mm hole lens might

be the one. The larger decrease in beam size for the second experiment is due to the initial beam size defined by the gate size, which is 1 mm throughout the second experiment. Looking at the heat maps of the beam size for the 3 mm Einzel lens, the compression of the beam on the membrane is noticeable for the second experiment (Figure 7b). A further decrease in the gate hole size, thus reducing the initial beam size, could improve the size of the electron beam at the membrane surface.

The increase in the size of the Einzel lens can influence not only the size of the electron beam but also the electron beam current value at the anode (anode current). The larger the Einzel lens, the higher the anode current (Figure 8); however, a type of saturation occurs after $a = 2$ mm, and even the current value drops for the 3 mm Einzel lens. For the second experiment, the values are smaller because of the screening of the beam by the gate electrode. Figure 8 represents the electron beam current at the anode $I_A$ normalized by the initial current $I_0$. The increase in the anode current along with a decrease in the spot size results in an increase in the brightness of the electron beam focused by the square Einzel lens, which is good for its use in the MEMS electron microscope.

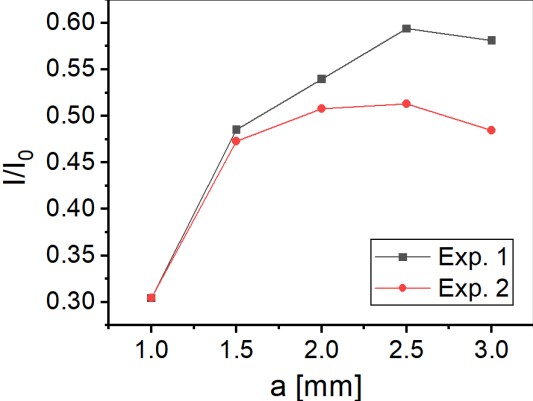

**Figure 8.** Normalized anode current as a function of the Einzel lens size.

The last parameter that was analyzed when studying the work of the developed microcolumn was the focusing voltage. During the calculation of the smallest beam diameter, a focusing voltage was recorded for every simulated model. It appears that the focusing voltage increases linearly with the size of the Einzel lens (Figure 9).

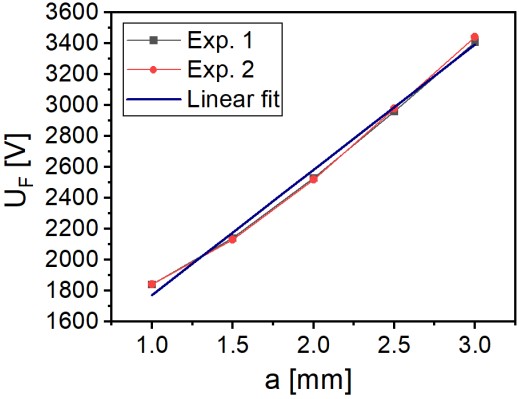

**Figure 9.** Best focus voltage as a function of Einzel lens size.

The larger the Einzel lens size, the higher voltage must be supplied to the focusing electrode. This parameter is important in terms of power supply design. Miniaturized high-voltage power supplies are difficult to design and implement. The most important of all voltages is the cathode voltage, which must be stable and precise to define the energy of the beam. The miniature MEMS electron microscope is designed to work with voltages up to 5 kV (cathode voltage). If the 3 mm Einzel lens was used to focus the electron beam with an energy of 2 keV, the focusing voltage is greater than −3.4 kV, which is 70% higher than the cathode voltage. When $U_C$ = −5 kV, $U_F$ = −8.5 kV, and that is a voltage that can lead to the electrical breakdown between the Einzel lens electrodes ($U_3$ and $U_5$ = 0 V). Therefore, a compromise must be determined in terms of the size of the electron beam and the focusing voltage.

### 3.2. Sharp Silicon/CNT Cathode

The results presented show that the best way to focus the electron beam is to increase the size of the square Einzel lens and limit the size of the initial electron beam by the gate electrode. However, the limitation of the size of the electron beam by the gate limits the electron beam current that reaches the anode. A better way to limit the initial size of the electron beam is to limit the size of the emitter. During research on the MEMS electron microscope, a sharp silicon/CNT cathode was developed [14], which meets these conditions. The cathode consists of a sharp silicon protrusion 200 μm high with a tip < 10 μm. The protrusion is covered by a CNT layer, which slightly enlarges the tip. To model such a cathode, a 200 μm protrusion was modeled on the flat surface of the cathode. Due to the resolution of the SIMION 3D program, the tip of the protrusion was set as a 40 μm× 40 μm square. From this platform, 50,000 electrons were emitted randomly and uniformly over the area. Similarly, the starting angles and electron energies were also randomly chosen.

Using such a defined cathode and a smaller initial electron beam, two experiments were conducted. At first, the simulations of the last models were repeated, with gate size 1 mm and Einzel lens size from 1 to 3 mm, every 0.5 mm. Calculations were made to estimate the beam diameter at the anode with a resolution of 0.1 mm and at the membrane with a resolution of 0.01 mm, as described earlier (Figure 10).

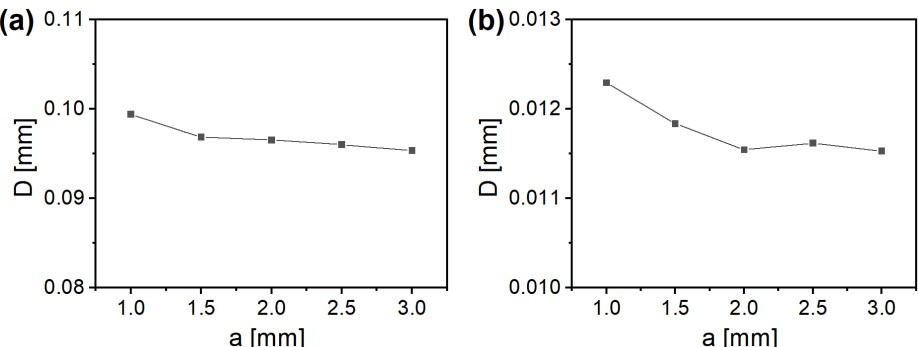

**Figure 10.** Electron beam size as a function of Einzel lens size: (**a**) at the anode; (**b**) at the membrane.

Analyzing the results of the calculations performed for the anode with 0.1 mm resolution, a strong decrease in the beam diameter is observed when the emission area is limited to 40 μm × 40 μm (Figure 10a). For the 3 mm Einzel lens, 1 mm gate electrode, the beam diameter is $D$ = 0.161 mm when using a 1 mm × 1 mm emitter and $D$ = 0.095 mm when using a 40 μm × 40 μm emitter. However, all estimated values for the anode are just below the resolution of the calculations, so conclusions cannot be drawn based on those results. Although, when calculations were made for the area of the membrane (with a

resolution of 0.01 mm), a strong peak in the center of the membrane was observed (Figure 11).

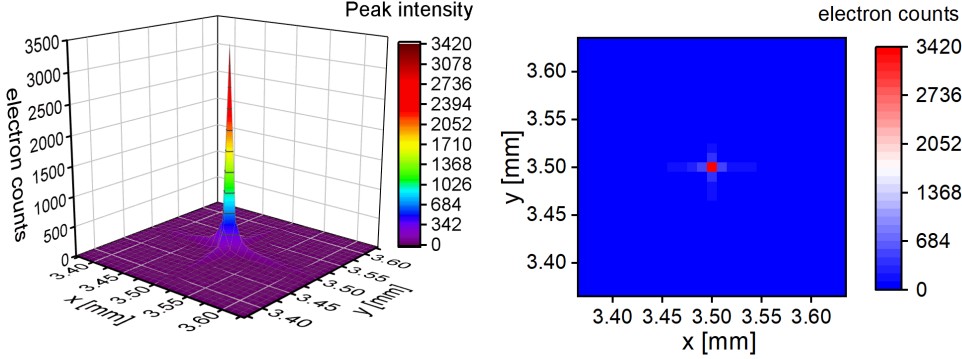

**Figure 11.** Three-dimensional plot and heat map for the best-focused beam with the 3 mm Einzel lens observed at the membrane.

The real values of the beam diameter for the sharp silicon/CNT emitter are presented in Figure 10b. Here, a decrease in the beam diameter is also observed, and saturation over 2 mm is visible. Hence, to focus a smaller initial electron beam, a 2 mm Einzel lens can be used.

Similar results to previous are visible when analyzing the currents at the anode and the membrane (Figure 12). However, the normalized anode current ($I_A/I_0$) is much higher than for the flat emitter, which means that the electrons emitted from a smaller area are not screened by the gate electrode (Figure 12, $I_A$). Furthermore, the saturation of the current is also observed. It seems that a 2 mm Einzel lens passes through the electrons that are later focused at the anode, which is consistent with the beam diameter calculations.

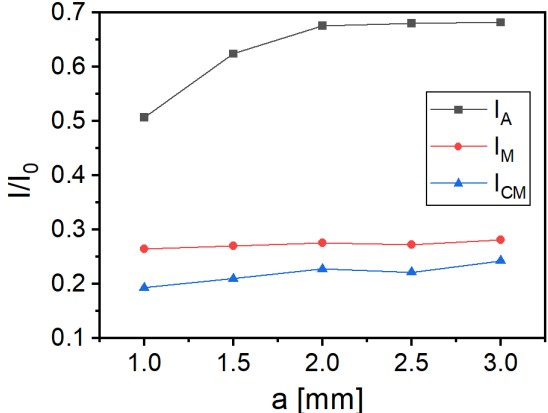

**Figure 12.** Normalized current as a function of Einzel lens size: $I_A$—normalized anode current ($I_A/I_0$), $I_M$—normalized membrane current ($I_M/I_0$), $I_{CM}$—normalized current at the center of the membrane ($I_{CM}/I_M$).

The results of the normalized current that reached the membrane ($I_M/I_0$) are quite different (Figure 12, $I_M$). It seems that despite the increase in the overall anode current, the membrane current is rather constant and does not depend on the size of the Einzel lens. However, when calculations were made of the number of electrons in the center square of the membrane and the resulted value was divided by the total number of electrons that hit the membrane ($I_{CM}/I_M$—plot 3), a slight increase is observed, which means that with a larger Einzel lens size, the focusing of the electron beam is better.

The focusing voltage increases linearly with the increasing size of the Einzel lens. Furthermore, the values are higher than for the 1 mm × 1 mm cathode (Figure 13), making the use of the 3 mm Einzel lens more problematic in terms of focusing voltage. The best focus voltage for a 3 mm lens is $U_F$ = 3590 V, which translates to 79.5% higher voltage than the cathode voltage, compared to 70% for the flat cathode. However, for a 2mm Einzel lens the $U_F$ = 2700 V, and this value (35% higher than the cathode voltage) is more acceptable in terms of steering voltages.

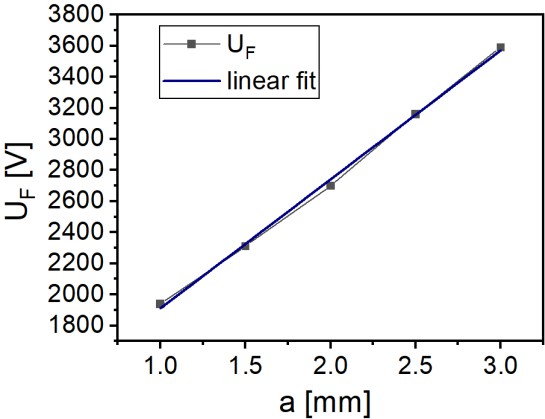

**Figure 13.** Best focus voltage as a function of Einzel lens size for sharp silicon/CNT cathode.

The second experiment utilizes the same model previously studied, with a sharp protrusion emitter and a 1 mm gate. However, the first and third Einzel lens electrodes are shifted by 100 μm in the *x* and *y* directions, respectively (Figure 14). We wanted to check how the Einzel lens will work when the electrodes are non-axial. We determined that our fabrication method allows us to align the electrodes with a precision of 100 μm. We performed the same simulations and sought the best focusing parameters.

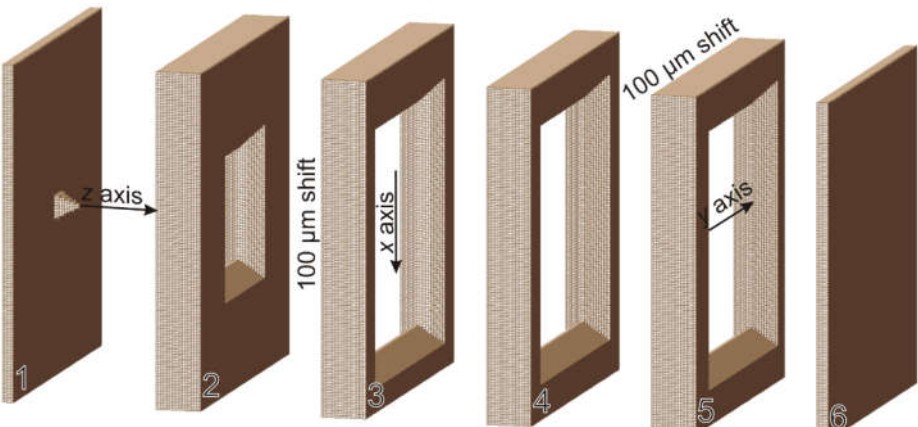

**Figure 14.** Scheme of the non-axial microcolumn.

The results are very promising. The size of the electron beam is larger only by a few percent (2.62% for 1.5 mm Eiznel lens) and for the 3 mm Einzel lens the beam is even smaller than for the axial microcolumn, which can be a calculation error (Figure 15a). Such little defocus observed in the non-axial microcolumn might be the result of a rather large size of the holes in relation to the electrode shift. The 100 μm shift is only a few percent of the hole size, and such a shift does not distort the electric field in a way that leads to de-

focus of the electron beam. This is very good information because high-precision alignment of the microcolumn is not needed in the proposed design of the miniature MEMS electron microscope. However, due to the shift of the Einzel lens electrodes, the center of the spot moves away from the center of the membrane when the larger Einzel lens is calculated (Figure 15b). This can be adjusted by the octupole scanning system.

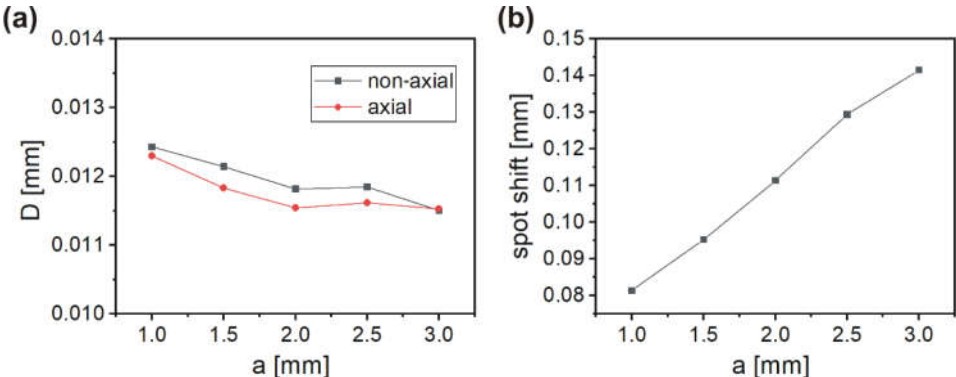

**Figure 15.** Results of non-axial microcolumn experiment: (**a**) electron beam size as a function of the Einzel lens size; (**b**) electron beam spot shift as a function of Einzel lens size.

## 4. Conclusions

The structure of the developed miniature MEMS electron microscope was modeled and simulated using SIMION 3D v.7.0 software. A six-electrode system was defined, including a cathode, gate, Einzel lens, and anode. The gate and Einzel lens electrodes had square holes, which is consistent with the design of the microscope. The electrodes were at a distance of $h$ = 1.1 mm, which is the thickness of the glass spacer used in the manufactured model.

The working principles and focusing parameters of the designed microcolumn were investigated using two realistic electron beams, which were consistent with the CNT cathodes fabricated and used in earlier research. Both beams incorporated 50,000 electrons randomly and uniformly distributed over the emission area. The start angles and energies are also randomized to conform to the real electron beam. The beams had different emission areas: the first is a flat emission layer 1 mm × 1 mm, which resembled the first realizations of the CNT cathode, the second is 40 μm × 40 μm area atop a 200 μm protrusion, which was similar to the latest sharp silicon/CNT cathode used in experiments.

All experiments carried out aimed to determine the spot diameter of the electron beam, the electron beam current at the anode, and the best focus voltage. In all experiments, a decrease in electron beam spot diameter was observed when the size of the Einzel lens increased. This is in opposition with studies on circular microcolumns, where smaller apertures (<100 μm) are used to focus the electron beam. Using small circular apertures, paraxial electrons are used to form a probe at the sample. The beam is focused in a small circular spot, and no additional electrons are present. In the presented solution, using the square Einzel lens, the best-focused spot is shaped like a cross with the most intensive beam in the center. It means that despite the presence of a high brightness electron beam spot, which can be used to generate imaging signal from the sample, additional electrons are also present, which contributes to a background signal. The background signal will lower the contrast of the image and can introduce artifacts. However, better focusing of the electron beam using a larger Einzel lens increases the electron beam spot brightness, which can result in better quality images. Moreover, the use of a larger Einzel lens increases the electron beam current that flows to the anode. This increases the brightness of the spot further, and that is also good information concerning the use of a square Einzel lens for the fabrication of a miniature MEMS electron microscope.

Better focusing can be achieved by limiting the initial electron beam size. This can be performed by decreasing the size of the gate hole. However, this decreases the electron beam current at the anode. The best way to limit the initial electron beam is to make the emitter size as small as possible. This observation is in agreement with the results of others. Conventional microcolumns consist of field emission cathodes in the form of sharp tungsten tips or single CNT, where the tip radius is less than 10 nm. Presented results for sharp silicon/CNT cathode confirm that limiting the size of the emitter has a positive influence on electron beam focusing using the square Einzel lens. It also increases the electron beam current at the anode.

Using large Einzel lenses improves the focusing capability of the microcolumn; however, the best focus voltage applied at the focusing electrode must be high. This factor must be taken into account when choosing an optimal microcolumn design. The single-lens consists of three electrodes. Two outer electrodes are at the ground state, and only the middle electrode (focusing electrode) is at high potential. The higher the focusing voltage, the higher the risk of electrical breakdown between the electrodes. That is why a compromise must be achieved between electron beam spots, hence the Einzel lens size and the focusing voltage.

The square Einzel lens has an advantage compared to conventional circular microcolumns, which was mentioned in the Introduction. The fabrication technology of square holes in silicon is easier and cheaper than that of circular holes. However, the simulations carried out revealed another advantage, which is the alignment of the electrodes in the microcolumn. In conventional microcolumns, where small apertures are used, the alignment must be very precise in order to achieve the best possible focusing, which further complicates the fabrication technology. When large square holes are used, the misalignment of 100 μm is acceptable. The results presented show that for two electrodes shifted away from the axis of the microcolumn by 100 μm, a 2.62% increase in the beam spot diameter is observed. The shift of the electrodes does not influence the electron beam diameter, but rather the displacement of the electron beam spot center at the membrane, and that can be adjusted by the octupole scanning system.

In conclusion, analyzing the results of modeling and simulation of the model of a miniaturized MEMS electron microscope, a final design of a microcolumn can be defined as a structure with a sharp emitter (with the smallest emission area possible), a gate electrode with 1 mm square hole, and Einzel lens with 2 mm square hole. However, more calculations should be performed to see if the gate electrode is optimal for this solution.

**Funding:** This research was funded by the National Science Centre Poland, project number UMO-2016/21/B/ST7/02216 and The APC was funded by the statutory fund of the Department of Microsystems, Wroclaw University of Science and Technology.

**Institutional Review Board Statement:** Not applicable.

**Data Availability Statement:** Data is available over request.

**Conflicts of Interest:** The author declare no conflicts of interest.

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
