# Peer review of "Design of an Einzel Lens with Square Cross-Section"

_electronics, doi:10.3390/electronics10192338_

Round 1

Reviewer 1 Report

Author of the manuscript entitled “Design of an Einzel lens with square cross-section” presents an original work of development of miniature MEMS electron microscope equipped with a miniature MEMS high-vacuum micropump. According to the author, such design makes this device the first stand-alone miniature electron optical device to operate without an external high-vacuum chamber.

In the Introduction Author report on the researches and different approaches already proposed in order to fabricate microcolumns using different techniques and different materials. The new approach is presented that include a production of square holes in the electrodes that build the microcolumn. For this purpose, the anisotropic silicon etching as a manufacturing technology was chosen. Number of papers reporting the tested particular elements of the designed microscope have already been published.

This article presents the results of simulations concerning the design of the square Einzel lens. The simulations concern only the Einzel lens were modeled and simulated and the author proposed a  final design of a microcolumn.

The manuscript is well organized; introduction provide sufficient background and include all relevant references for the topic. Research design is appropriate and well explained. A scientific approach and research methods were used.  Results are clearly presented and conclusions as well.

The acceptance of the manuscript is suggested with minor corrections. The author should write in the indefinite gender and not in the first person plural (for example, instead of “we made” it should be “it was made”). This needs to be corrected in several places in the text.

The acceptance of the manuscript is suggested, with those minor corrections.

Author Response

Thank you for your review and for your comment. Of course, the corrections proposed by you are most accurate, and I have rewritten the text using indefinite gender. Corrections made in the text are colored red. Due to suggestions of the second reviewer, some corrections to the text were also made. I hope that the corrected manuscript will also be accepted by you.

Reviewer 2 Report

In this manuscript, the author presents the design of a microcolumn with square holes in the electrodes. The dependence of the electron beam spot size on other parameters is investigated and discussed. The structural organization and method description are clear. As highlighted in the title, the novelty of this work is the square holes in the electrodes. However, I don't see sufficient discussions of the advantages of the square holes in the introduction part, with only one sentence "This is chosen because we wanted the microcolumn to be as simple as possible." in line 58 mentioned. 

To highlight the novelty and improve the clarity of this work, I suggest the author:

1) draw 3D structures show the "conventional hole" and "square hole". Indicate clearly the geometry parameter and dimensions;

2) list the advantages and disadvantages of this idea (square hole lens) with some other shapes. Describe clear why you choose the square shape and are there any other researchers working on this in the literature. 

3) please indicate the Si3N4 film in the schematic in Figure 1. 

4) Please also include the top view of the Einzel lens. a 3D structure would be great.

5) In Fig.8, please describe more clearly what is the difference between exp 1 and exp 2. A schematic could be used to describe the difference. 

6) Fig.14 is not mentioned in the text. 

7) what are "axial" and "nonaxial", how much is the non-axial displacement? is that 100 um? Please explain a bit why there is not much difference in Fig.14. Please add the schematic and explain your fundamental understanding. 

8) try to narrow down your conclusion parts, and put more explanations for each Figure, not just listing the results. You have to show the fundamental physics to explain the results. 

Author Response

In this manuscript, the author presents the design of a microcolumn with square holes in the electrodes. The dependence of the electron beam spot size on other parameters is investigated and discussed. The structural organization and method description are clear. As highlighted in the title, the novelty of this work is the square holes in the electrodes. However, I don't see sufficient discussions of the advantages of the square holes in the introduction part, with only one sentence "This is chosen because we wanted the microcolumn to be as simple as possible." in line 58 mentioned.

Reply: The discussion of the advantages of the square holes in the introduction part is poor, because it is hard to discuss parameters of the square Einzel lens before knowing the results of the simulation. However, you have a valid point, that I should state some hypothetical questions in the introduction part, and improve my work to your next comments, to be clear what is the purpose of this my work. I will also add discussion about advantages of the square Einzel lens in the conclusion part of the manuscript.

To highlight the novelty and improve the clarity of this work, I suggest the author:

1) draw 3D structures show the "conventional hole" and "square hole". Indicate clearly the geometry parameter and dimensions;

Reply: I have rendered the 3D structure of the model from SIMION 3D software. As I used the square holes in the electrodes from the start of the project, this is the only image that I was able to present. I hope that it is more clear to the cross-section presented previously.

2) list the advantages and disadvantages of this idea (square hole lens) with some other shapes. Describe clear why you choose the square shape and are there any other researchers working on this in the literature.

Reply: The only advantage of the square holes in the electrostatic lenses, to the best of my knowledge, is the simple and not expensive fabrication technology. To fabricate circular holes used by others, an isotropic etching of the silicon must be made. This can be done by reactive ion etching (RIE) or deep reacting ion etching (DRIE) processes, which are expensive and use expensive tools. The square holes can be done by anisotropic silicon etching in alkali solutions like 10 M KOH solution that we use in our laboratory. The accuracy of this process is limited mostly by photolithography resolution and quality; however, it uses very cheap ingredients and simple laboratory equipment. I have not found any square Einzel lens in the literature, presumably because the axially symmetric circular holes are best to use to form electrostatic lenses where the quality of the electrostatic field generated by the lens should not be disturbed.

In this article, my intentions are not to compete with 'conventional structures' but to try to answer the question whether the square Einzel lens could be used as an alternative.

I have clarified this in the introduction section of my manuscript.

3) please indicate the Si3N4 film in the schematic in Figure 1.

Reply: I have added the indication of the membrane to Fig. 1.

4) Please also include the top view of the Einzel lens. a 3D structure would be great.

Reply: Since the manuscript already has 15 figures adding another one might be too much. Hopefully, the corrected Fig. 2 can be the image of the Einzel lens, as it is a part of the microcolumn. 

5) In Fig.8, please describe more clearly what is the difference between exp 1 and exp 2. A schematic could be used to describe the difference.

Reply: All Figs. 6, 8, and 9, represent different results of two experiments described for flat CNT cathode at the beginning of the section. In exp. 1, all electrode sizes are changed, and in exp. 2, the gate size is constant (a = 1 mm) and only the size of the Einzel lens (three electrodes) size is changed. Since the manuscript already has 15 figures adding another one might be too much.

I hope that a more clear description and indication of the particular dimensions in Fig. 2 will be sufficient to clarify the text.

6) Fig.14 is not mentioned in the text.

Reply: I have added the Fig. 14 reference to the text.

7) what are "axial" and "nonaxial", how much is the non-axial displacement? is that 100 um? Please explain a bit why there is not much difference in Fig.14. Please add the schematic and explain your fundamental understanding.

Reply: The axial microcolumn has all the electrode centers aligned on the axis. Non-axial microcolumns have electrodes, the centers of which are not on the axis. In my case, the first and third Einzel lens electrodes (electrodes 3 and 5) are shifted by 100 µm of axis in the x and y directions. I have added a 3D scheme to clarify the arrangement as a Fig. 14 and rearranged the results to fit Fig. 15.

The little difference in the beam size of the axial and non-axial microcolumns could be the result of a rather large hole size in relation to the electrode’s shift. The 100 µm shift is only a few percent of the hole size, and this leads not to defocusing of the electron beam but rather to shifting of the electron beam spot center, as shown in Fig. 15b.

An explanation was added to the text.

8) try to narrow down your conclusion parts, and put more explanations for each Figure, not just listing the results. You have to show the fundamental physics to explain the results.

Reply: I have discussed the results of the particular figures in the Results sections of the manuscript, and therefore in the conclusions I am just recalling the main results. However, I have rewritten the conclusions and added discussion on the results.

Round 2

Reviewer 2 Report

The authors have addressed all my comments. I agree with the acceptance of this paper.